# Mapping the Energetic Costs of Free-Swimming Gilthead Sea Bream (*Sparus aurata*), a Key Species in European Marine Aquaculture

**DOI:** 10.3390/biology10121357

**Published:** 2021-12-20

**Authors:** Sébastien Alfonso, Walter Zupa, Maria Teresa Spedicato, Giuseppe Lembo, Pierluigi Carbonara

**Affiliations:** COISPA Tecnologia and Ricerca, Experimental Station for the Study of Sea Resources, Via dei Trulli 18-20, 70126 Bari, Italy; zupa@coispa.it (W.Z.); spedicato@coispa.it (M.T.S.); lembo@coispa.it (G.L.); carbonara@coispa.it (P.C.)

**Keywords:** acoustic telemetry, sea bream, MO_2_, *U*
_crit_, electromyogram, metabolic cost, welfare monitoring, aquaculture

## Abstract

**Simple Summary:**

Assessment of the energetic costs of different living activities is of primary interest among fish biologists. However, assessing energy expenditure in free-swimming fish is challenging owing to the difficulty of performing such measurements in the field. Therefore, the use of implant fish with sensors that transmit signals that serve as a proxy for energy expenditure is a promising method to counter these limitations, allowing remote monitoring in tagged fish. The aim of this study was to correlate the acceleration recorded by the tag with the activities of the red and white muscles and the oxygen consumption rate (MO_2_), which could serve as a proxy for energy expenditure, in gilthead sea bream (*Sparus aurata*), a key species in European marine aquaculture. The acceleration recorded by the tag was successfully correlated with MO_2_. Additionally, through electromyographic analyses, we determined the activities of the red and white muscles, which are indicative of the contributions of aerobic and anaerobic metabolisms during swimming. Finally, the tag implantation did not affect the swimming performance, metabolic traits, and swimming efficiency of the sea bream. By obtaining insights into both aerobic and anaerobic metabolisms, sensor mapping with physiological indicators may be useful for the purposes of aquaculture health/welfare remote monitoring of gilthead sea bream.

**Abstract:**

Measurement of metabolic rates provides a valuable proxy for the energetic costs of different living activities. However, such measurements are not easy to perform in free-swimming fish. Therefore, mapping acceleration from accelerometer tags with oxygen consumption rates (MO_2_) is a promising method to counter these limitations and could represent a tool for remotely estimating MO_2_ in aquaculture environments. In this study, we monitored the swimming performance and MO_2_ of 79 gilthead sea bream (*Sparus aurata*; weight range, 219–971 g) during a critical swimming test. Among all the fish challenged, 27 were implanted with electromyography (EMG) electrodes, and 27 were implanted with accelerometer tags to monitor the activation pattern of the red/white muscles during swimming. Additionally, we correlated the acceleration recorded by the tag with the MO_2_. Overall, we found no significant differences in swimming performance, metabolic traits, and swimming efficiency between the tagged and untagged fish. The acceleration recorded by the tag was successfully correlated with MO_2_. Additionally, through EMG analyses, we determined the activities of the red and white muscles, which are indicative of the contributions of aerobic and anaerobic metabolisms until reaching critical swimming speed. By obtaining insights into both aerobic and anaerobic metabolisms, sensor mapping with physiological data may be useful for the purposes of aquaculture health/welfare remote monitoring of the gilthead sea bream, a key species in European marine aquaculture.

## 1. Introduction

Measurement of metabolic rates in animals, including fish, is of primary interest because it provides a valuable proxy for the activity-dependent energetic costs of different living activities [1,2,3]. In recent decades, ecophysiological studies have been conducted on the basis of the estimation of the following metabolic traits: standard metabolic rate (SMR), maximum metabolic rate (MMR), and aerobic scope (AS) [1,2]. The SMR represents the minimum amount of oxygen needed by fish to support their aerobic metabolic rate [2]. MMR refers to the maximum rate of aerobic metabolism of an animal and is, therefore, associated with the maximum rate at which oxygen can be transported from the environment to tissue mitochondria [3]. Typically, these two metabolic traits are measured during a critical swimming test (*U*_crit_). The SMR can be estimated on the basis of the relationship between oxygen consumption rate (MO_2_) and swimming speed, extrapolating the value at the speed of zero. The MMR is generally recorded at the *U*_crit_ value during the critical swimming test [1,2]. Finally, the numerical difference between MMR and SMR describes the absolute AS, which quantifies the amount of oxygen that can be consumed to support all physiological and locomotive activities, such as migration, feeding, or reproduction [4,5].

However, the metabolic traits, based on the estimation of MO_2_, do not account for the energetic costs that can be imputed to anaerobic metabolism. Owing to its less efficient metabolic process, anaerobic metabolism is not adapted to sustain swimming over long periods but rather only for brief and intense swimming bursts such as escape behaviour [6,7]. Greater insights into the species-specific activation proportion of the aerobic and anaerobic metabolisms during swimming activities may be achieved through electromyographic (EMG) analyses [8]. Briefly, the EMG signal is a biomedical signal that is used to measure the electrical currents generated in muscle cells during muscle contraction. EMG signals could also be used in studies of fish physiology to further investigate and describe the activation patterns of the red and white muscles during swimming, and thus, serve as a proxy for the energetic expenditures related to aerobic and anaerobic metabolisms [9,10,11]. Indeed, swimming at a slow speed is sustained by the slow contraction of red muscle fibres, but when the velocity increases, the faster fibres of the white muscle tend to be recruited more, with a species-specific mechanism of activation [12,13]. Although measurements of MO_2_ and muscular activity are needed to carefully estimate fish energy expenditure, they are not easy to apply in fish in the wild and aquaculture conditions [1].

In the last decades, different bio-sensing techniques such as the use of an accelerometer or EMG tags have been developed for estimating the energetic costs of the behaviours of free-swimming fish, both in the wild and in aquaculture environments [14,15,16,17,18,19]. These technologies have been proven to be sensitive for remote monitoring of health and welfare in farmed fish [14,15,16,20,21,22,23,24] and could be promising tools in the context of precision livestock farming [25,26,27,28]. However, the use of such devices in free-swimming fish for estimating energetic costs related to swimming, including those in aquaculture conditions, requires mapping with physiological data such as MO_2_ [29,30,31,32,33]. As the MO_2_ is known to vary as a function of different biotic and abiotic factors (e.g., species, life stage, stress state, temperature, water quality, and oxygen concentration), the inference of MO_2_ using accelerometer tags needs an accurate calibration for a specific species, at a given size and temperature, before it can be used in aquaculture environments [1,2].

The gilthead sea bream (*Sparus aurata* Linnaeus, 1758) plays an ecological key role and is of primary importance for European marine aquaculture [34]. However, little information is known about the swimming performance and energy expenditure of this species [35,36,37,38,39,40]. The aim of this study was to correlate the acceleration recorded by accelerometer tags with MO_2_ and the activity patterns of the red and white muscles to later estimate the energetic costs of different life activities in free-swimming tagged fish. To this end, we first assessed the swimming performance of sea bream, ranging from 219 to 971 g in weight, in critical swimming tests (*U*_crit_), and then estimated the metabolic traits (SMR, MMR, and absolute AS) and swimming efficiency (minimum cost of transport: COT_min_ and optimal swimming speed: *U*_opt_). A sub-sample of fish was implanted with accelerometer tags, and the acceleration recorded by the tag was correlated with the MO_2_ recorded during the trial, for use as a proxy for the energetic costs related to aerobic metabolism. Swimming performance, metabolic traits, and swimming efficiency were compared between the tagged and untagged individuals to assess the possible effects of such implantation on these variables. Finally, EMG analyses were performed to determine the activation patterns of the red and white muscles during the *U*_crit_ trials, thereby monitoring the contributions of aerobic and anaerobic metabolisms at different swimming activity levels. Overall, the estimation of MO_2_ and the activities of the red and white muscles on the basis of the measurement of acceleration from the tags during fish swimming could benefit to the use of the sensor for remote health and welfare monitoring of gilthead sea bream in aquaculture environments.

## 2. Materials and Methods

All the experiments in this study were conducted in accordance with the Italian national legislation (D. lgs. 26/2014) and EU recommendations (Directive 2010/63/EU) on fish welfare, with the authorisation of the Italian Health Ministry, under protocol code 665/2016-PR and 838/2019-PR.

### 2.1. Fish-Holding Conditions

Gilthead sea bream were purchased from Panittica Italia SRL (Torre Canne, Italia) and then kept in our facility (Bari, Italy) for 6 weeks for acclimation before proceeding with the experimental procedures. The fish were reared in 1.2-m^3^ circular fiberglass tanks in a flow-through system with a marine water input of 150 L/h (35 PSU). The stocking density was approximately 15 kg/m^3^, oxygen saturation was maintained at >80%, and temperature was kept at 18 ± 1 °C. A constant light regimen was maintained during the entire experimental period (12-h light/12-h dark). The fish were fed with commercial food (Marine 3P, Skretting, Stavanger, Norway) at 1% of the body mass.

### 2.2. Critical Swimming Speed Tests (U_crit_), Estimation of Metabolic Traits, and Cost of Transport

Critical swimming speed tests (*U*_crit_) were conducted using 30- and 90-L swim tunnel respirometer Loligo Systems (Viborg, Denmark; https://www.loligosystems.com/swim-tunnel-respirometer-3 (accessed on 20 December 2021); models SW10150 and SW10200, respectively), in accordance with the manufacturer’s instruction regarding fish body mass. The trials were controlled using the DAQ-M device (No. AR12500, Loligo Systems, Viborg, Denmark; https://www.loligosystems.com/daq-m-instrument (accessed on 20 December 2021)), whereby a honeycomb screen was installed at the entrance of the swimming chamber to minimise turbulence and ensure that the water had a uniform velocity profile. The water flow speed was measured prior to the experiment by using the flowmeter Flowtherm NT (Höntzsch, Germany), while the oxygen concentration was measured using a polymer optical fibre oxygen probe (Loligo Systems) inserted in the swimming chamber. The oxygen probe was connected to WITROX oxygen instrument 1 (No. OX11800; fitted with a high-accuracy temperature sensor; Loligo Systems; https://www.loligosystems.com/witrox-1-oxygen-meter-for-mini-sensors-1-x-o-1-x-temp (accessed on 20 December 2021)) to record oxygen variations during the trials. Finally, the swimming chambers were housed in a buffer tank to guarantee accurate temperature control (18 ± 1 °C).

The fish were fasted 24 h before starting the swimming trial to ensure a post-absorptive state [41] and lightly anesthetised (stage I: reduced motion and breathing) using a 30-mg/L hydroalcoholic clove oil solution (Erbofarmosan, Bari, Italy) for the morphometric measurements [9,42]. After introduction into the swimming chamber, the fish were left undisturbed for at least 30 min before the induction of a slow swimming velocity of 0.1 m/s. This velocity was maintained for at least 90 min for fish acclimatation before the start of the swimming trial. The swimming trial began when the MO_2_ reached a constant low plateau at a water speed of 0.1 m/s [43]. The *U*_crit_ trials were conducted by imposing a swimming speed ramp (0.1 m/s) at constant time intervals (10 min) until fatigue was reached (i.e., when the caudal fin touched the back grid of the swimming chamber for at least 5 s [44]). Each water speed step was composed of three periods: a 5-min period of ‘flushing’, 2-min period of ‘waiting’, and 3-min period of ‘MO_2_ measurement’. Briefly, during the flushing period, the flush pump actively pumped water from the ambient temperature bath and into the respirometer. After 5 min, the flush pumping was stopped with a short waiting period before starting the measurement period. The waiting period was necessary to account for the lag in the system response, resulting in a non-linear oxygen curve. During the measurement period, the flush pump was off, and the chamber was closed. During the MO_2_ measurement period, the oxygen concentration in the swim tunnel water was recorded every second, and MO_2_ was automatically calculated using the AutoResp v.2.3 software (Loligo Systems, Viborg), from a linear decrease in O_2_ concentration inside the chamber, measured using the appropriate constants for oxygen solubility in seawater (salinity, temperature, and barometric pressure).

The *U*_crit_ values were estimated using Brett’s method [45] and corrected for solid blocking effects [46,47]. Absolute *U*_crit_ values (m/s) accounted for fish size and were displayed as relative *U*_crit_ (BL/s). The MO_2_ was fitted as a function of water speed during the swimming trials, and the model predictions were used for estimating the values of the metabolic variables SMR, MMR, and AS (see details in Section 2.5). The cost of transport (COT) was fitted as the function of swimming speed (U, km/h) in accordance with the formula developed by Zupa et al. [9] as follows:(1)$$\mathrm{COT}=\frac{a\;e^{\mathrm{bU}}}{U},$$

The COT function, characterised by a *U* shape, has a minimum (COT_min_) corresponding with the *U*_opt_ (i.e., best swimming efficiency) [48]. *U*_opt_ was calculated as the first derivative function of the COT equation as follows:(2)$$U_{\mathrm{opt}}=\frac{1}{b},$$

In the present study, 79 sea breams were challenged in a *U*_crit_ trial (from 219 to 971 g). Of the 79 fish, 25 were tested, as described earlier, without any additional device, to trace their baseline swimming performance and MO_2_ (later called untagged), 27 were challenged using hard-wire EMG for modelling the red/white muscle activity (later called EMG), and 27 sea breams were challenged using an accelerometer tag (later called tagged) to correlate the acceleration data from the tag with the MO_2_ during the trial. The morphometric data of the sea breams tested in the study are reported in Table 1 for each condition, and the protocol is displayed in Figure 1.

### 2.3. EMG Analysis: Monitoring of the Red and White Muscle Activities during the U_crit_ Trial

The sub-group of 27 sea breams was randomly selected to assess the activation patterns of the red and white muscles by performing EMG analyses during the *U*_crit_ trials (Table 1). The experimental procedure was similar to that described in the previous section, except that the surgical procedure was performed before the swimming trial to insert electrode wires into red and white muscles (Figure 1). During the surgery, the gills were continuously irrigated with anaesthetic solution (40-mg/L hydroalcoholic clove oil solution), as described elsewhere [9]. Briefly, two pairs of plastic-coated stainless-steel wire electrodes (California Fine Wire Company, Grover Beach, CA, USA; 0.1-mm thin and 1-m long) were surgically implanted subcutaneously using syringe needles (NIPRO, Zaventem, Belgium) in both the lateral red and white muscles (in the same location but approximately 1 cm below the surface for white muscle) [9]. The two electrodes of each pair were placed at least 10 mm apart from one another to avoid potential contact during contraction. The wires were sutured to the left side of the body to minimise entanglement during the *U*_crit_ trial.

During the swimming trials, EMG signals for the red and white muscles were sampled with an analogue-to-digital interface DAQCardAI-16E-4 (National Instrument, Austin, TX, USA; Appendix A) at 500 data points per second and digitally converted using the National Instruments USB-6009 device (sample rate: 48 kS/s), amplified with Grass P511 preamplifiers (Grass Technologies, West Warwick, RI, USA), filtered, and root-mean-square (RMS) averaged using the LabVIEW software (SignalExpress, National Instruments, USA) [9]. The RMS EMG values were averaged for each swimming velocity step of the trial and accounted for by the maximal RMS values recorded for each fish during the trial; that is, the maximal value reached during the trial was 100% for all fish. Similarly, for all speed steps, each water speed was accounted for by the *U*_crit_ value (100%) reached by the fish and displayed as a percentage.

After the *U*_crit_ trial, the fish recovered at a slow speed (0.1 m/s) for around 2 h before the wires were removed from the fish, under similar anaesthesia conditions to the surgical procedure. An antibiotic injection (sodic-ampicillin–cloxacillin, 1 mg/kg) was given to avoid any health issues. All the challenged fish recovered well from the surgery, and no mortality related to the surgery occurred. Among the 27 fish challenged, four and six individuals were excluded from the data analysis of the red and white muscles, respectively, because of wrong positioning of the hard-wires or signal transmission issues occurring during the trial, resulting in the analyses of 23 and 21 sea breams for determining the red and white muscle activities, respectively.

### 2.4. Implantation of Accelerometer Tags: Recording Fish Acceleration during the U_crit_ Trial

A subgroup of 27 fish (Table 1) was randomly selected to correlate the swimming activity, measured by accelerometer tags in the tail-beat mode, with the MO_2_ during the *U*_crit_ trials. As for the implantation of electrodes, the fish were fasted 24 h before the surgery and then anesthetised using a 30-mg/L hydroalcoholic clove oil solution. Briefly, the acoustic transmitter VEMCO V9AP (AMIRIX Systems Inc., Bedford, NS, Canada; length, 43 mm; weight, 6.1 g in air and 3.3 g in water) was implanted in the body cavity, close to the urogenital opening (nearest to the caudal fin), through a 1.5-cm incision, and carefully sutured, as described elsewhere [49] (Figure 2). After the surgery, an antibiotic injection (sodic-ampicillin-cloxacillin, 1 mg/kg) was administered, and the fish were left undisturbed in a separate tank (1.2-m^3^ circular fiberglass tank) for 5 days to recover before being challenged in the *U*_crit_ trial.

At a sampling rate of 10 Hz, the acoustic transmitters were programmed to record the acceleration over two axes (*x*- and *z*-axis), removing the backward/forward acceleration (*y*-axis; Figure 2) [50]. The accelerometer tag transmitted the tag ID and the coded values corresponding to the acceleration every 30 s on average (from 15 to 45 s). The tag returned an 8-bit value that represented the root mean square (RMS) acceleration. The values ranged from 0 to 255 arbitrary units (AUs) and can be converted into acceleration (m/s^2^) using the following equation [acceleration (m/s^2^) = 0.01955(*x*), where *x* is the adimensional value returned by tags), resulting from the contribution of two axes (vertical and lateral directions of movement). The acceleration data were stored in memories of submergible acoustic receivers (Vemco VR2W; AMIRIX Systems Inc., Bedford, NS, Canada) located in the swimming chamber until further processing. At the end of the trial, the data were extracted from the acoustic receiver using the VUE software (AMIRIX Systems Inc., Bedford, NS, Canada) and were averaged for each swimming speed step before processing for correlation with the MO_2_ values.

At the end of the swimming trial, the fish was left to recover at a slow speed (0.1 m/s) for 1 h before the acoustic transmitter was removed, under a similar protocol as that for electrode removal. All the challenged fish recovered well from the surgery, and no mortality related to the surgery occurred. Among the 27 fish challenged, four were excluded from the data analysis because of the loss of signals from the acoustic transmitters (end of battery life of the transmitter), resulting in 23 fish included in the analysis.

### 2.5. Statistical Analyses

All analyses were performed using the R version 4.0.4 software [51] and at the 95% level of significance. Data are presented as mean ± SE unless otherwise specified.

First, swimming performances (*U*_crit_) were compared between the three conditions (untagged, tagged, and EMG) using the Kruskal–Wallis test, followed by Dunn’s test adjusted by Bonferroni correction. Subsequently, swimming performances (*U*_crit_) were evaluated as a function of fish mass and conditions by using an analysis of variance (ANOVA) and interactions of both factors. As both the condition and the interaction between mass and condition were not significant (Appendix A), they were removed from the ANOVA. Thus, only the results of the linear regression analysis between *U*_crit_ and mass are presented. As the swimming performance was different from the other groups, the fish in the EMG group were excluded from the analysis, modelling of MO_2_ during the trial, and estimation of metabolic traits to avoid underestimation of the swimming performance and metabolic traits of the fish.

The modelling of MO_2_ with swimming speed during the trial was tested using a linear, exponential, or logistic model before the trial. The best model was selected on the basis of the Akaike information criterion, and only the best model (i.e., logistic) is presented below. In more detail, the MO_2_ of the sea bream during the *U*_crit_ trials was modelled as a function of swimming speed using self-starting non-linear least squares logistic models (SSlogis) for each fish. The SSlogis model is based on the following formula:(3)$$y=\frac{A_{\mathrm{sym}}}{1+e^{\frac{x_{\mathrm{mid}}-x}{\mathrm{scal}}}},$$
where *A*_sym_ is a numeric parameter representing the asymptote, *x*_mid_ is a parameter representing the *x* value at the inflection point of the curve, *y* = *A*_sym_/2, scal is a scale parameter for the *x*-axis (swimming speed), and *x* represents the speed of the water during trial.

Model predictions were used for estimating the values of the metabolic variables SMR, MMR, and absolute AS. For each fish, the SMR was estimated by extrapolating the MO_2_ at speed = 0 [52], and the MMR was estimated using the maximum oxygen consumption rate displayed at the *U*_crit_ value [3]. The numerical difference between MMR and SMR indicates absolute AS. Finally, linear regressions were applied for SMR, MMR, AS, COT_min_, and *U*_opt_ as a function of fish mass to evaluate the relationships of metabolic traits and swimming efficiency with the mass of sea bream. Additionally, all metabolic traits (SMR, MMR, and AS) and the COT_min_ and *U*_opt_ were compared between the tagged and untagged fish using the Student *t* test to investigate the effect of tagging on these variables.

Concerning the muscle activation pattern, the EMG signals from the red and white muscles were fitted using nonlinear models. In particular, the red muscle EMG data were modelled using an SSlogis model, whereas the white muscle was modelled using an exponential model. The exponential model was based on the following formula:(4)$$y=\;\alpha \;e^{\beta x},$$

To determine the ‘break point’ between the activation of the red and white muscles, that is, when the increment of the white muscle became greater than that of the red muscle during the *U*_crit_ trial [53], the following equation was solved:(5)$$f^{\prime }(x_{\mathrm{EMG}\;\mathrm{white}})>f^{\prime }(x_{\mathrm{EMG}\;\mathrm{red}}),$$
where *x*_EMG white_ and *x*_EMG red_ refer to the models describing the activation pattern for EMG white and red muscles, respectively (i.e., exponential and sigmoid, respectively).

Finally, to correlate the swimming activity recorded by the tag with the MO_2_, two modelling steps were performed. In the first step, the MO_2_ was fitted as a function of the acceleration recorded by the tag using the SSlogis model for each fish (Appendix A). The parameters of the model (i.e., *A*_sym_, *x*_mid_, and scal) were extracted for each fish and evaluated as a function of fish mass using linear regression (Appendix A). In the second step, a unique SSlogis model was fitted, accounting for all fish. For this model, the parameters were adapted depending on the relationship between the parameters (i.e., obtained for the first step for each fish) and fish mass. In step 1, *A*_sym_ was the only variable parameter according to fish mass as follows: *A*_sym_ = 696.8 − 0.34 × mass (*p* < 0.001; *R*^2^ = 0.6; Appendix A). Thus, for step 2, the *A*_sym_ parameter was changed from fixed to the equation displayed above, accounting for fish mass, whereas the other parameters were kept fixed (as they were not influenced by mass) and estimated using the classic iteration method.

## 3. Results

### 3.1. Critical Swimming Speed (U_crit_), Estimation of the Metabolic Traits, and COT

Overall, both the absolute and relative *U*_crit_ values differed depending on the fish condition (i.e., untagged, tagged, and EMG; *p* < 0.001 and *p* = 0.04 for absolute and relative *U*_crit_ values, respectively). The fish implanted with EMG wires displayed lower absolute *U*_crit_ values than the two other conditions (*p* < 0.05 for both; 0.92 ± 0.07 for EMG vs. 1.03 ± 0.1 and 0.99 ± 0.11 m/s for the untagged and tagged conditions, respectively; Appendix A). Additionally, the fish implanted with EMG wires also differed in their relative *U*_crit_ values in comparison with the untagged fish, but not with the tagged fish (3.19 ± 0.57, 3.00 ± 0.78, and 2.77 ± 0.52 BL/s for the untagged, tagged, and EMG conditions, respectively). Thus, the fish implanted with EMG wires were excluded from the analysis of the link of swimming performance and mass to MO_2_ during the trial and estimation of the metabolic rates of sea bream. For both the absolute and relative *U*_crit_ values, the performances negatively correlated with the fish mass regardless of the tagging condition (*p* < 0.001 for both absolute and relative *U*_crit_; Appendix A); larger sea bream displayed lower swimming performances. The swimming performances were found to be similar between the untagged and tagged fish (Appendix A).

During the *U*_crit_ trial, the MO_2_ progressively increased until reaching the asymptote, following a sigmoid model in the sea bream (Figure 3). The sigmoid models well explained the MO_2_ during *U*_crit_ in the sea bream (*p* < 0.001 for all fish; *R*^2^ ranked from 0.89–1.0; see Appendix A for the statistical details). As shown in Figure 3, larger fish reached the plateau sooner with lower MO_2_ values.

On the basis of the model, the SMR value was extrapolated from the MO_2_ value at a speed of 0, and the MMR was extrapolated from the MO_2_ value at the *U*_crit_ trial for each fish. The SMR is comparable between fish, regardless of the fish mass (89.14 ± 28.17 mgO_2_/kg/h; *p* > 0.05; Appendix A). On the contrary, the MMR decreased with the increase in fish mass (*y* = 731.36 − 0.33 × *x*; *p* < 0.001; *R*^2^ = 0.58; Appendix A). As a consequence, the absolute aerobic scope also decreased with the increase in fish mass (*y* = 624.83 − 0.35 × *x*; *p* < 0.001; *R*^2^ = 0.52; *p* < 0.001; Appendix A), resulting in a lower absolute aerobic scope in the larger fish. Additionally, all the metabolic traits (SMR, MMR, and absolute AS) were not significantly different between the tagged and untagged sea breams (*p* > 0.05 for all the traits; Appendix A).

The COT_min_ was 147.20 ± 21.25 mgO_2_/kg/km in the sea bream, resulting in the best swimming efficiency at a swimming speed of *U*_opt_ = 2.40 ± 0.54 km/h (Figure 4; Appendix A). In addition, neither COT_min_ and *U*_opt_ were found to be affected by fish mass (*p* > 0.05 for both) or tagging condition (*p* > 0.05 for both; Appendix A; Appendix A).

### 3.2. Red and White Muscle Activation Patterns during the Critical Swimming Test

The activation pattern of the red muscle was described using a sigmoid function (Table 2; *p* < 0.05), similarly to the MO_2_ pattern during the *U*_crit_ trials (Figure 5). On the contrary, the white muscle showed very low activation at low speeds, whereas its contribution increased according to the exponential pattern near the *U*_crit_ value (*p* < 0.05; Table 2; Figure 5).

To determine the ‘break point’ between the activation of the red and white muscles [53], that is, when the increment of the white muscle becomes greater than the red muscle during the *U*_crit_ trial, Equation (6) was solved, where *x*_EMG white_ and *x*_EMG red_ refer to the functions that describe the activation patterns of the EMG white and red muscles, respectively. According to the functions that describe the activation of the white and red muscles (i.e., exponential and sigmoid, respectively), the derivative functions are as follows:(6)$$f^{\prime }(x_{\mathrm{EMG}\;\mathrm{white}})=0.0122516\;e^{0.31114x},$$
and:(7)$$f^{\prime }(x_{\mathrm{EMG}\;\mathrm{red}})=\frac{4.63108\;e^{0.0457603\left(41.604-x\right)}}{{\left(1+e^{0.0457603\left(41.604-x\right)}\right)}^{2}},$$

As shown in Figure 5, the break point was located at 65% of the *U*_crit_ value for sea bream. After this threshold, anaerobic metabolism begins to progressively compensate for the slowdown of aerobic metabolism to fuel the swimming of sea bream.

### 3.3. Correlation of Acceleration Recorded by the Tags with the MO_2_

In the fish implanted with accelerometer tag, the fish acceleration increased as a function of the increase of water speed following an exponential pattern during the *U*_crit_ trial (*p* < 0.05 for all fish; Figure 6; Appendix A for the statistical details). On the basis of this model, the swimming activity appeared to increase faster in the larger fish than in the smaller fish (Figure 6); thus, the correlation of swimming activity with MO_2_ also included mass as an explanatory factor.

The MO_2_ is explained as a function of the swimming activity recorded by the tag using a sigmoid model (Table 3; *R*^2^ = 0.82). In this model, the *A*_sym_ parameter was replaced with a linear equation accounting for fish mass (see Section 2.5 for details). According to this model, the MO_2_ can be estimated with the following formula:(8)$${\mathrm{MO}}_{2}=\frac{696.8-0.34*x_{2}}{1+e^{\frac{x_{\mathrm{mid}}-x_{1}}{\mathrm{scal}}}},$$
where *x*_1_ is the swimming activity recorded by the transmitter and *x*_2_ is the fish mass (in grams). *x*_mid_ and scal were estimated as constants depending on the fish mass, 34.4 ± 1.0 and 19.3 ± 1.3 g, respectively (Table 3).

According to this model, for low acceleration values displayed by the tag, the MO_2_ increased but increased faster in the smaller fish than in the larger fish. The MO_2_ asymptote (*A*_sym_) was reached sooner by the larger fish, which means that the MO_2_ asymptote reached was lower than that reached by the smaller fish (Figure 7).

## 4. Discussion

Despite the ecological key role of gilthead sea bream and its primary importance in the Mediterranean marine aquaculture [34], only little information is known about the swimming performances and metabolisms of this species, which are both useful for conservation policy making and health and welfare monitoring in the aquaculture environment. In this study, we (i) provide the baseline swimming performance for differently sized sea breams, MO_2_, and activation patterns of red and while muscles, all of which contribute to the estimation of metabolic costs related to swimming. We also (ii) correlated the acceleration recorded by the accelerometer tags with MO_2_ for use as a proxy for the energetic costs related to aerobic metabolism in free-swimming tagged fish.

Fish species exhibit a wide range of specialisation in swimming activity, which is the result of varying body and fin shapes, species-specific arrangement of muscle fibres, and different contractile and metabolic properties [12,54,55]. The gilthead sea bream, as a species of the Sparidae family, has an oblong, tall, and compressed body. The species is characterised by a carangiform swimming locomotion, in which the thrust is produced by the rear third of the body length, while the anterior part is relatively inflexible, and a rigid caudal fin that accommodates the fish’s turning and accelerating abilities [54,56]. In the present study, the values of relative *U*_crit_ ranged from 2.02 to 4.79 BL/s and were consistent with the values reported in previous studies for this species [35,38]. In addition, the swimming performance (either absolute or relative *U*_crit_) of the sea bream during the critical swimming speed test was reduced with the increase in fish body mass. The decrease in relative *U*_crit_ was consistent with the overall pattern observed in fish and has already been reported for sea bream [35,57]. Fish implanted with EMG hardwires displayed lower swimming performance than the untagged and tagged fish. This was mainly due to bathing in clove oil, which has a relatively long half-life in plasma (approximately 14 h), and/or due to stress induced by the surgical operation a few hours before the *U*_crit_ testing. To account for this difference in swimming performance between the fish under different conditions (untagged, tagged, and EMG), when further analysing the muscle activity pattern, the EMG signal was studied as a function of the relative percentage of *U*_crit_ (Figure 6; instead of the water velocity step), and EMG implanted fish were not used to model the MO_2_ during the *U*_crit_ test and to estimate the metabolic traits. However, the accelerometer tag insertion in the body cavity (5 days before the test) did not trigger any significant changes in the swimming performance of the tested fish, showing that (i) the tag implantation (5 days before the trial) did not impact the swimming performance of the fish, which suggests the low invasiveness of the surgery, as also shown by previous studies [38,49,58]. Additionally, it showed that (ii) the reliability of the measurements performed on the fish implanted with a transmitter can be expected to be similar to that of the measurements performed on the fish without a transmitter, which supports their use as a tool for remote health/welfare monitoring in aquaculture conditions.

In fish, the MO_2_ is overall modelled as a function of swimming speed using linear, exponential, or power functions [10,59,60,61]. In this study, the MO_2_ was, however, fitted using a sigmoid function to show the best fit with the data. Indeed, the sigmoid function describes the initial exponential pattern of the consumption rate and the slowdown near the *U*_crit_. This relationship between MO_2_ and swimming speed was already observed in sea bream [40,43], in similar species such as sea bass (*Dicentrarchus Labrax*) [9], or in more distant species such as rainbow trout (*Oncorhynchus mykiss*) [32] or chinook salmon (*Oncorhynchus tshawytscha*) [53]. These differences in MO_2_ modelling may be due to the different protocol methodologies, because our protocol and those cited earlier were based on a shorter time step than those cited previously. Nevertheless, the estimation of metabolic traits (SMR and MMR, respectively), MO_2_ at speed 0, and *U*_crit_, based on the prediction from the sigmoid model, provides results consistent with those reported in the literature on sea bream. For instance, Steinhausen et al. measured SMR values of 131 ± 24.3 and 96.5 ± 26.4 mgO_2_/kg/h at *t* = 20 °C, during forced and spontaneous swimming, respectively. In the present study, we estimated the SMR to be constant across fish sizes at 89.1 ± 28.2 mgO_2_/kg/h, slightly lower than those measured by Steinhausen et al. [62] regardless of the method, which could be explained by the different experimental temperatures (18 °C vs. 20 °C). However, other studies reported relatively higher values for sea bream SMR for a similar temperature range of 18–20 °C (208 mgO_2_/kg/h at 18 °C and 204 or 209 mgO_2_/kg/h at 20 °C) [63,64]; this may be due to the variability related to the experimental design used to measure SMR [52]. Concerning the MMR, we estimated an average value of 526.2 ± 97 mgO_2_/kg/h for the sea breams tested, which is higher than the values reported by Martos-Sitcha et al., who estimated the MMR to be between 355 and 450 mgO_2_/kg/h at a temperature of 24–25 °C [43]. Besides the effects of size and/or temperature on metabolic variables, a three-fold intraspecific change in SMR and MMR values between individuals has already been observed [3,65], with a higher variation observed for MMR. Finally, as for *U*_crit_, the MMR was reduced with the increase in fish size, resulting in a decrease in aerobic scope with the increase in fish size. Thus, smaller sea bream with greater AS have a greater amount of oxygen that can be used for routine activities, to invest in physiological processes, and/or to cope with stress. Additionally, no significant differences in metabolic traits were found between the tagged and untagged fish, indicating that overall, tagging did not induce stress that could impact metabolic rates in this species. This is consistent with previous works that measured other end points (e.g., growth performances, cortisol levels) [49,59,66]. Additionally, both the optimal swimming speed (*U*_opt_ = 2.40 ± 0.54 km/h) and associated COT_min_ (147.20 ± 21.25 mgO_2_/kg/km) measured in this study were consistent with those reported in the literature on the species [37,38,39] and were similar between the tagged and untagged fish. This confirms the recent results of Arechavala-Lopez et al. [38], who showed that the tagging procedure had no effect on the swimming efficiency of sea bream (at least 5 days after the tagging procedure), and supports the use of such tags as non-invasive tools for estimating the energy expenditure in sea bream.

Contrary to the recent results of Arechavala-Lopez et al. [38] in sea bream, our results showed a correlation between the MO_2_ values and the acceleration values recorded by the tags. The main explanations for such differences between the two studies are the location of the transmitters and the algorithms associated with the measurements. The accelerometer tags used by Arechavala-Lopez et al. [38] were implanted in the body cavity, above the pelvic girdle, whereas we implanted tags in the body cavity close to the urogenital opening (nearest to the caudal fin), which better highlights the energetic costs related to swimming [67]. Additionally, the tags used by Arechavala-Lopez et al. [38] recorded the three dimensions of movement (i.e., *x*-, *y*-, and *z*-axes), whereas our algorithm only recorded the *x*- and *z*-axes, removing the backward/forward acceleration (*y*-axis). Removing the *y*-axis from the algorithm calculation allows for accounting only for the undulation movements associated to the movements of the tail (‘tail-beat beat’ mode); thus, more reliable estimates of energetic costs could be obtained [29,32,67]. This highlights that the tag implantation location and the associated algorithms measuring acceleration are crucial for such correlations and for estimating energetic expenditure. In our model, the sigmoid function was applied to fit the MO_2_ data as a function of fish swimming activity and mass. This function allows for describing the initial exponential increase in MO_2_ as the function of fish acceleration and then the slow-down to reach the asymptote near the MMR (following the MO_2_ trend during the *U*_crit_ trials). The asymptote of the model was reached earlier (i.e., with a lower MO_2_ value) for larger fish than for smaller ones, which is consistent with the size-related metabolic variables. Thus, the model accounts for the decrease in MMR due to the increase in the mass of the sea bream and, thus, provides a reliable correlation between MO_2_ and the acceleration measured by tags for this species. It is important to note that for estimating MO_2_ based on tag acceleration in free swimming fish, the oxygen level of water needs to be higher than the limiting O_2_ saturation (LOS) of the species. LOS is defined as the threshold level at which regulatory mechanisms are no longer sufficient to maintain O_2_ consumption without compromising any physiological function [36,68]. In gilthead sea bream, Remen et al. [36] observed that the LOS increased exponentially with temperature, from 17% to 35% of the O_2_ saturation, when passing from a temperature of 12 °C to 20 °C. Based on the LOS-temperature model developed by Remen et al. [36], we advise the use of our calibration model for free-swimming fish in water with O_2_ saturation >28% at a temperature of 18 °C. Additionally, later calibrations should account for temperature to improve the accuracy of MO_2_ estimates based on transmitter outputs [29] or could be retrieved from the literature [36]. Inference of the MO_2_ is valuable because in addition to the acceleration data recorded by tag, it can be used as a proxy for the energetic costs related to different activities of free-swimming fish in the wild and in the aquaculture context [69,70]. However, metabolic costs cannot be inferred only by the measurement of MO_2_, as swimming may also be supported by anaerobic metabolism, including during high-intensity swimming bursts [7]. This can be observed for swimming activity values > 100 AU. The swimming activity still increased, whereas the MO_2_ remained stable; this indicates that for high values recorded by tags, anaerobic metabolism is the contributing factor that fuels swimming.

In this sense, greater insights into the species-specific activation of aerobic and anaerobic metabolisms during swimming may be achieved through electromyographic analyses [8]. In the present study, the sigmoid model used to describe the data obtained by the hard-wire EMG signals from the red muscle was consistent with the MO_2_ pattern of sea bream during the *U*_crit_ trials (aerobic metabolism) [40]. Indeed, the slower activation of the oxidative red muscular fibres at higher swimming speeds is supported by the progressively decreasing rates of oxygen uptake close to the critical swimming speed (curve plateau), limited by the diffusion velocity of gases at the tissue and cellular levels [71]. While the red muscle supports swimming with aerobic metabolism for the full range of sub-critical swimming speeds, the contribution of the white muscle at lower speeds is negligible until 65% of the *U*_crit_ (corresponding to tag values of approximately 77 AU; Figure 5). Thereafter, a progressive increase in white muscle recruitment is observed until *U*_crit_ is reached, following a typical exponential pattern. Owing to the capability of white fibres to contract and relax faster than red muscle fibres, fish can reach higher tailbeat frequencies and swimming speeds, which can help them cope with intense energy requirements (e.g., escape and predation). These metabolic costs must be accounted for the high values recorded by the tags to properly estimate the energy expenditure related to swimming. Thus, coupled with the correlation of the signals from the accelerometer tags with the MO_2_, this study provides valuable data for use as a proxy for energy expenditure related to swimming activities in tagged gilthead sea bream [69,70].

## 5. Conclusions

In conclusion, this study provides the baseline swimming performance (from 2.02 to 4.79 BL/s) for differently sized gilthead sea bream (from 219 to 971 g), MO_2_, and activation patterns of the red and white muscles during swimming. During the swimming trial, the MO_2_ showed a sigmoid pattern with an exponential increase during the first velocity steps before reaching asymptote at the end of the trial. Estimate of SMR was 89.1 ± 28.2 mgO_2_/kg/h, and MMR was on average 526.2 ± 97 mgO_2_/kg/h at 18 °C, resulting in a decreasing absolute aerobic scope with an increase of fish body mass. The optimal swimming speed *U*_opt_ was found to be 2.40 ± 0.54 km/h, and was associated to a COT_min_ of 147.20 ± 21.25 mgO_2_/kg/km in gilthead sea bream. Red muscle (aerobic) showed a similar activity pattern to MO_2_, while white muscle (anaerobic) showed an exponential pattern, negligible contribution until 65% of the *U*_crit_, and then a strong increase at the end of the trial. Additionally, we showed that tag implantation did not affect swimming performance, swimming efficiency, and metabolic traits, supporting the use of such tags for health and welfare monitoring purposes. Finally, the acceleration recorded by the accelerometer tag during trial exponentially increased with swimming speed regardless of fish mass but increased faster in larger fish. The correlation of acceleration recorded by the tag with the MO_2_ and mass (i.e., sigmoid model) provides a valuable proxy for the aerobic energetic costs of sea bream. Overall, acoustic telemetry offers the possibility of obtaining a vast amount of data covering a long observation period, while reducing the number of specimens used and minimising fish-handling disturbances. Thus, such sensors can be implanted in free-swimming fish to estimate MO_2_ variations in response to environmental changes or aquaculture practices [14,15,16,20,21,22,23]. Such a study could bring new perspectives for health and welfare monitoring in aquaculture environments by remotely providing a proxy for energy expenditure related to different rearing practices for this species in the framework of precision livestock farming.

## Figures and Tables

**Figure 1 biology-10-01357-f001:**
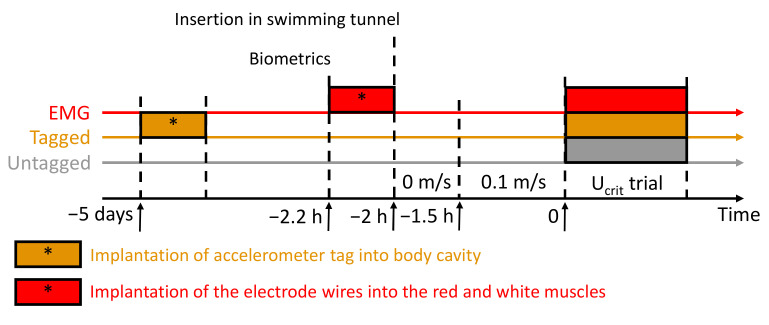
Experimental protocol regarding the critical swimming speed (*U*_crit_) trial for the untagged (grey timeline), tagged (orange timeline), and EMG (red timeline) gilthead sea bream (*Sparus aurata*). Negative time refers to the start of the *U*_crit_ trial (0). The time of implantation of the accelerometer tag or electrode wire into the red and white muscles is indicated with an asterisk (*).

**Figure 2 biology-10-01357-f002:**
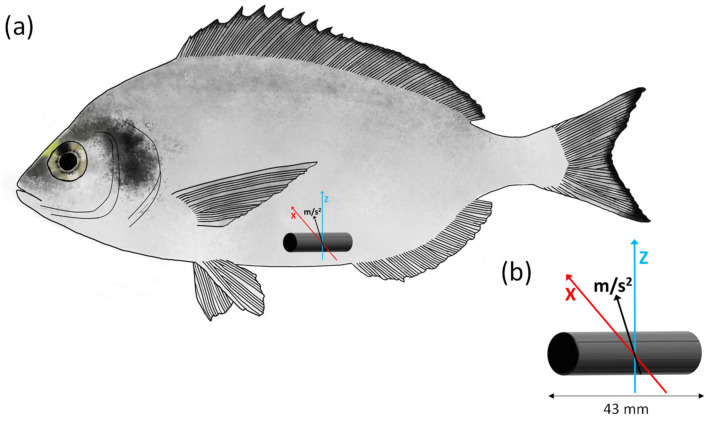
(**a**) Location and position of the V9AP acoustic transmitter in the body cavity of the gilt-head sea bream (*Sparus aurata*). (**b**) The tag is programmed to measure the acceleration (m/s^2^) over two axes (*x*-axis, red and *z*-axis, blue).

**Figure 3 biology-10-01357-f003:**
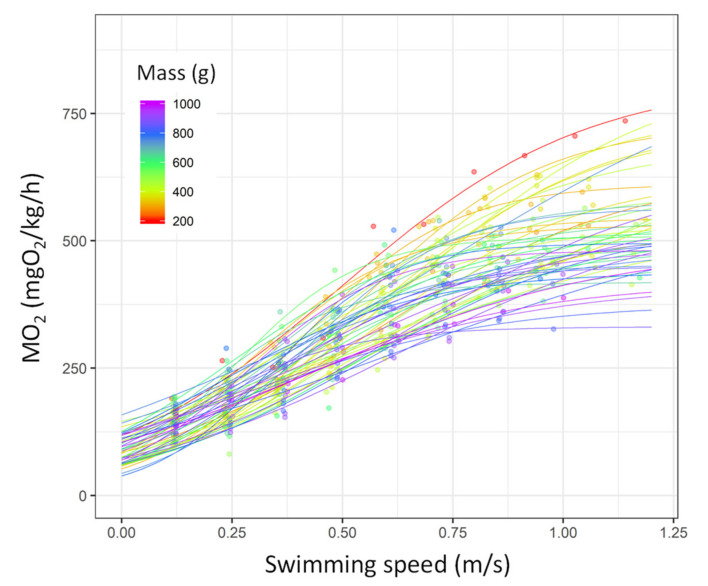
Oxygen consumption rate (MO_2_, mgO_2_/kg/h) as a function of swimming speed in gilthead sea bream (*Sparus aurata*; *n* = 52 fish). Each line represents the relationship between the two variables for each single fish based on the logistic models (the statistical details of the model parameters for each fish are shown in Appendix A). Each dot represents a value obtained during the *U*_crit_ trial. The points and line colours refer to the mass of the fish (from 200 to 1000 g, from red to purple).

**Figure 4 biology-10-01357-f004:**
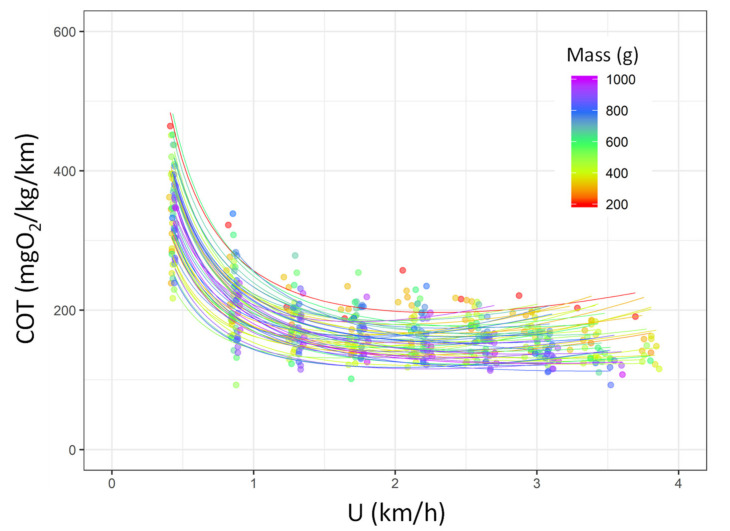
Cost of transport (COT, mgO_2_/kg/km) as a function of speed (km/h) during the *U*_crit_ trial of the gilthead sea bream (*Sparus aurata*; *n* = 52). Each line represents the relationship between the two variables for each single fish based on the logistic models. Appendix A shows the statistical details of the model parameters for each fish. Points and line colours refer to the mass of the fish (from 200 to 1000 g, from red to purple).

**Figure 5 biology-10-01357-f005:**
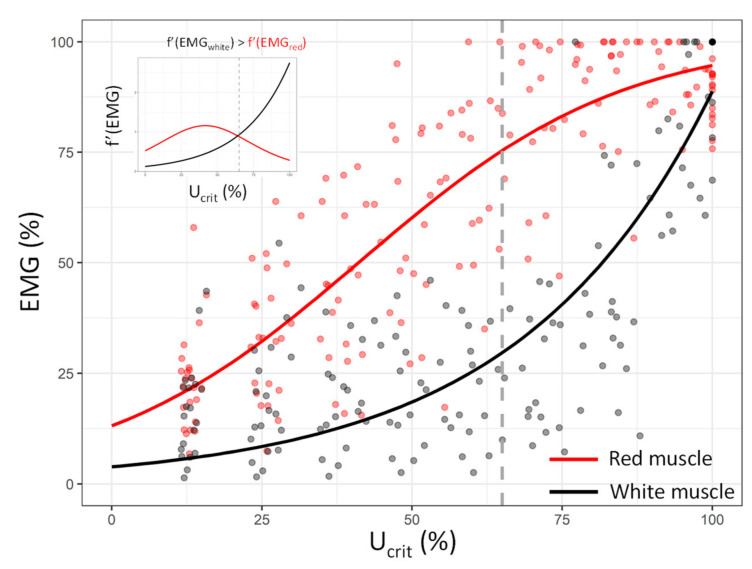
Electromyographic (EMG) levels (%) for the red muscle (red dots, *n* = 23 fish) and white muscle (black dots, *n* = 20 fish) as a function of *U*_crit_ (%) in gilthead sea bream (*Sparus aurata*). The red line indicates the sigmoid model fitting the EMG levels of the red muscle, and the black line indicates the exponential model fitting the EMG levels of the white muscle during the swimming trial (see statistical details in Table 2). The dashed grey line indicates the break point, where the increment of the white muscle becomes greater than the increment of the red muscle. The upper right of the figure shows the method used to determine the break point according to the derivative function (65% of the *U*_crit_ value; see the main text for details).

**Figure 6 biology-10-01357-f006:**
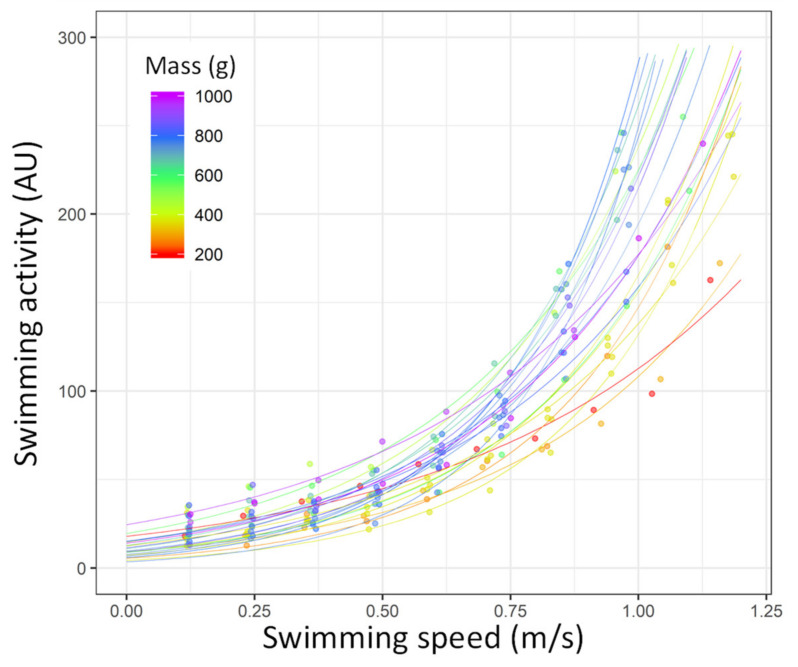
Swimming activity (arbitrary unit [AU]) recorded by accelerometer tags as a function of swimming speed during the *U*_crit_ trial (m/s) in gilthead sea bream (*Sparus aurata*; *n* = 23). Each line represents the relationship between the two variables for each single fish based on the exponential model (y = α × e^β^ × ^x^; the statistical details of—and β for each fish are shown in Appendix A). Each dot represents a value obtained during the *U*_crit_ trial. The points and line colours refer to the mass of the fish (from 200 to 1000 g, from red to purple).

**Figure 7 biology-10-01357-f007:**
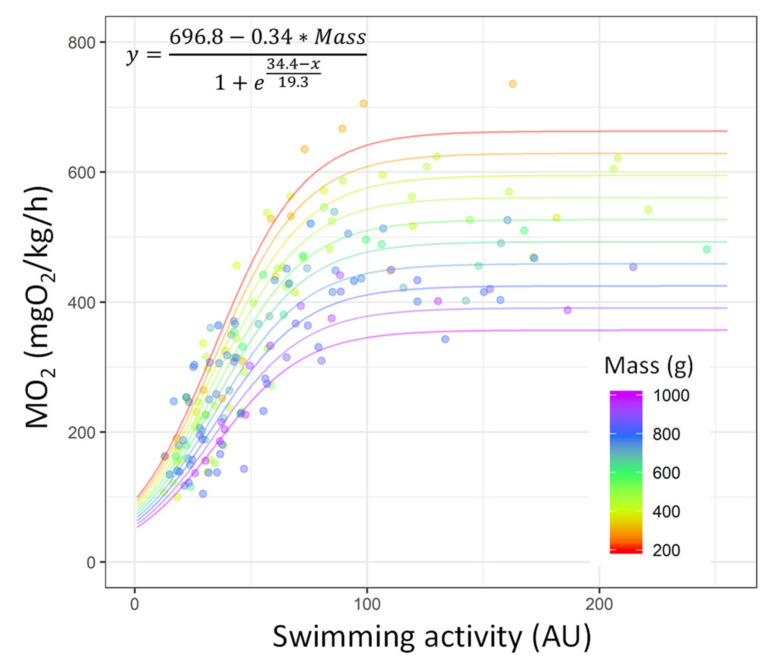
Correlation of the MO_2_ values (mgO_2_/kg/h) with the swimming activity (arbitrary unit (AU)) recorded by accelerometer tags in gilthead sea bream (*Sparus aurata*; *n* = 23). The lines are plotted on the basis of the calibration model predictions (*R*^2^ = 0.82). Each dot indicates a value obtained during the *U*_crit_ trial. The dots and line colours refer to the mass of the fish (from 200 to 1000 g, from red to purple).

**Table 1 biology-10-01357-t001:** Number and morphometric measurements (mass (g) and total length (TL; mm)) of the gilthead sea bream (*Sparus aurata*) challenged in the *U*_crit_ trials according to conditions (untagged, tagged, and EMG). The data presented are mean ± SD.

Condition	*n*	Mass (g)	Total Length (mm)
Untagged	25	590.39 ± 195.05	328.36 ± 41.78
Tagged	27	647.27 ± 224.85	342.93 ± 47.5
EMG	27	631.07 ± 205.67	340.77 ± 44.88
Total	79	623.73 ± 207.91	337.58 ± 44.74

**Table 2 biology-10-01357-t002:** Summary of the outputs of the models fitting the electromyographic (EMG) signal for the red (*n* = 23) and white muscles (*n* = 20) during the *U*_crit_ trials in gilthead sea bream (*Sparus aurata*). The associated *R*^2^ is 0.75 and 0.72 for the EMG of the red and white muscles, respectively.

**EMG Signal—Red Muscle**				
Parameter	Estimate	Std. error	*t* Value	*p* Value
*A* _sym_	101.203	4.600	22.001	<0.001
*x* _mid_	41.604	2.893	14.381	<0.001
scal	21.853	2.556	8.549	<0.001
**EMG Signal—White Muscle**				
Parameter	Estimate	Std. error	*t* Value	*p* Value
α	3.938	0.731	5.383	<0.001
β	0.031	0.002	15.467	<0.001

**Table 3 biology-10-01357-t003:** Outputs of the significant linear regression for the *A*_sym_ parameter (*p* < 0.001; *R*^2^ = 0.59), and for the model for the calibration of the MO_2_ as a function of the swimming activity recorded by the accelerometer tag (SSlogis, *p* < 0.001; *R*^2^ = 0.82) in gilthead sea bream (*Sparus aurata*; *n* = 23).

***A*_sym_—Linear Regression**				
Parameter	Estimate	Std. error	*t* Value	*p* Value
(intercept)	696.79	38.28	18.20	<0.001
Mass	−0.34	0.06	−5.74	<0.001
**Calibration model—SSlogis**				
Parameter	Estimate	Std. error	*t* value	*p* value
*x* _mid_	34.43	1.01	33.9	<0.001
scal	19.36	1.33	14.51	<0.001

## Data Availability

The raw data that support the findings of this study are available from the corresponding author upon reasonable request.

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
