# Peer review of "Mapping the Energetic Costs of Free-Swimming Gilthead Sea Bream (Sparus aurata), a Key Species in European Marine Aquaculture"

_biology, 2021, doi:10.3390/biology10121357_

Round 1

Reviewer 1 Report

Regarding this article, it seems to be a very novel and interesting topic for research work, since energy expenditure during swimming is an interesting parameter for fish biologists and also to be applied on research works on aquaculture. Acceleration tags are being used lately in fish research but there is a lack of knowledge about the relation of acceleration parameters with muscle activity or oxygen consumption.

The work supplements several other investigations covering the same topic and many of these papers appear in the references. Overall, the paper is well written and considerable thought has been put into interpreting the reported findings. Written English is good, and the manuscript is easy to read and understand. The structure and format are good, and the title and abstract give the necessary information about the work.

The introduction is good, emphasizing the principal subjects of the paper to present the hypothesis. Authors underline the importance of using sensors for remote health and welfare monitoring and with this study, they make a great advance on this subject.

The authors explained all the followed methodology in a correct and structured way. They used figures to accompany the text, making it easier to understand all the experimental designs and protocols. The results section is good on structure and content. The authors have taken care to support the text with figures and tables, specifying in the text in greater detail the data that are represented in the figures. The discussion is also good, and the authors show that they have a great background on the subject and know previous research about accelerometry on fish.

Due to the aforementioned, if the other reviewers and editor agree, I consider that the article should be ready to publish in Biology Journal.

Author Response

Regarding this article, it seems to be a very novel and interesting topic for research work, since energy expenditure during swimming is an interesting parameter for fish biologists and also to be applied on research works on aquaculture. Acceleration tags are being used lately in fish research but there is a lack of knowledge about the relation of acceleration parameters with muscle activity or oxygen consumption.

The work supplements several other investigations covering the same topic and many of these papers appear in the references. Overall, the paper is well written and considerable thought has been put into interpreting the reported findings. Written English is good, and the manuscript is easy to read and understand. The structure and format are good, and the title and abstract give the necessary information about the work.

The introduction is good, emphasizing the principal subjects of the paper to present the hypothesis. Authors underline the importance of using sensors for remote health and welfare monitoring and with this study, they make a great advance on this subject.

The authors explained all the followed methodology in a correct and structured way. They used figures to accompany the text, making it easier to understand all the experimental designs and protocols. The results section is good on structure and content. The authors have taken care to support the text with figures and tables, specifying in the text in greater detail the data that are represented in the figures. The discussion is also good, and the authors show that they have a great background on the subject and know previous research about accelerometry on fish.

Due to the aforementioned, if the other reviewers and editor agree, I consider that the article should be ready to publish in Biology Journal.

Response: We are delighted of very positive feedback from the reviewer regarding our manuscript. We would to thank him/her to take time for reviewing our manuscript before publication. To fix some concerns regarding the two other reviewers, the manuscript has been adapted. Please find the changes highlighted using the “Track Changes” function.

Reviewer 2 Report

Overview

In the manuscript Using acceleration recorded by accelerometer tags for estimating oxygen consumption rate and muscle activity of Gilthead Sea bream (Sparus aurata), the aim of this study was to correlate the acceleration recorded by a tag with the activity of red and white muscles, and the oxygen consumption rate (MO2), which could serve as a proxy of energy expenditure in Gilthead Sea bream (Sparus aurata), a key species of the European marine aquaculture. The theme is relevant and the manuscript is well written. The experimental design is consistent, robust and the statistics analyses can be appropriate to the approaches used. Suggestions for adjustments are few and are specified below. However, the authors were not convincing about the reasons for the study.  Considering that this study can help both for conservation policy and aquaculture health/welfare monitoring was not convincing. For conservation aspects, the authors could explore the great importance of this approach for studies evaluating swimming behaviors and capacities for migratory species that need to use fishways. To justify the use of results for aquaculture purposes, the authors should consider or propose the tolerance limits for the species (critical velocities, dissolved oxygen rates and saturation) in crop environments, if the species actually coexists with flow velocities, such as when using aerators or natural currents in net tanks. Thus, the study could recommend tolerant limits of dissolved oxygen/saturation rates and for current flows in cultivation environments. This may be useful for planning facilities in more favorable locations for future fish production projects.

In addition, there are specific corrections to make throughout the manuscript that are commented below.

Specific comments

Title

The tag, is a commercial product already used and was only a means to get the results. Thus, the objective of the study was not the development and testing of this brand, which does not justify including it in the title of the manuscript. It is noteworthy that other tests were performed, besides estimating the rate of oxygen consumption. So, I recommend the readjustment of the title.

Simple Summary

Line 14 - The study was conducted, so the verb is in the past.

Introduction

Consider aspects of behavior and swimming capacity of migratory species that use fishways. Consider values of tolerance limits (dissolved O2, Saturation %, current velocities, etc.) established for the rearing of the studied species.

Material and methods

An illustration of a schematic drawing would be very useful for understanding, containing the equipment used (swimming chambers), as well as the design of the studies carried out (acclimatization, biometrics, use of the tag, use of electrodes).

Lines 224, 125, 126, 127, 129, 130, 131, 132 and 133 - What models/part numbers are used? Quote them.

Results

Line 408 - Figure 6 - Keeping the formula in Figure 6, I do not consider it necessary to quote here, just include the value of R2 in Figure 6.

Conclusions

It should be improved by focusing on the main results of the study and its possible application both for conservation (behavior and swimming capacity) and for aquaculture (tolerant limits to be established).

Reviewer 3 Report

Reviewer Report

This is an interesting investigation that showed how to use accelerometer tags for estimating oxygen consumption rate and muscle activity of Gilthead Sea bream (Sparus aurata). The authors were able to correlate the acceleration recorded by tag with the fish MO2. This finding has a potential for high impact to the field. However, I have some major concerns

Major concerns:

  1. Fish size has been demonstrated to impact fish MO2. Giving the variable size range (219-971g) used in this investigation. How did the authors account for the differences in fish size?
  2. Handling stress is another variable that impact MO2 and swimming rates- how did the investigators account for this variable since not all fish were tag by implication the untagged fish were not subjected to the same stress as tagged fish.
  3. What was the reason for discarding the values from fish implanted with EMG-wires?

Round 2

Reviewer 2 Report

Dear Authors,
The authors responded to most of the suggestions made. However, most of the Conclusion was restricted to addressing the advantages of the tools used in the study (tag and acoustic telemetry). The authors should emphasize the numerical values found in this study (MO2 x Swimming Speed; COT x Ucrit; EMG x Ucrit; Swimming Activity x Swimming Speed), and not those other studies that can guarantee the welfare of the species, in production systems (aquaculture).

Author Response

The reviewer is right. We added all the information suggested by the reviewer in the conclusion of the revised version of the MS. Nevertheless, we keep some words regarding the use of such calibration for aquaculture purposes as perspectives. Once again, we are grateful to the reviewer for providing insightful comments that improved the manuscript.

Reviewer 3 Report

The authors have sufficiently addressed all my concerns.

Author Response

We are pleased to see that the reviewer's concerns are now fixed. Thank again for take time to review our manuscript.